# High-Throughput Functional Analysis of CFTR and Other Apically Localized Proteins in iPSC-Derived Human Intestinal Organoids

**DOI:** 10.3390/cells10123419

**Published:** 2021-12-04

**Authors:** Sunny Xia, Zoltán Bozóky, Michelle Di Paola, Onofrio Laselva, Saumel Ahmadi, Jia Xin Jiang, Amy L. Pitstick, Chong Jiang, Daniela Rotin, Christopher N. Mayhew, Nicola L. Jones, Christine E. Bear

**Affiliations:** 1Molecular Medicine, Hospital for Sick Children, 686 Bay St, Toronto, ON M5G 0A4, Canada; sunny.xia@mail.utoronto.ca (S.X.); zbozoky@providencehealth.bc.ca (Z.B.); onofrio.laselva@unifg.it (O.L.); jiaxinjanet.jiang@sickkids.ca (J.X.J.); 2Cell Biology, Hospital for Sick Children, Toronto, ON M5G 0A4, Canada; chong.jiang@sickkids.ca (C.J.); drotin@sickkids.ca (D.R.); nicola.jones@sickkids.ca (N.L.J.); 3Department of Physiology, University of Toronto, Toronto, ON M5S 1A8, Canada; michelle.dipaola@mail.utoronto.ca; 4Department of Medical and Surgical Sciences, University of Foggia, 71122 Foggia, Italy; 5Department of Neurology, Washington University in St. Louis, St. Louis, MO 63110, USA; saumel@wustl.edu; 6Division of Developmental Biology, Cincinnati Children’s Hospital Medical Center, Cincinnati, OH 45229, USA; amy.pitstick@cchmc.org (A.L.P.); christopher.mayhew@cchmc.org (C.N.M.); 7Department of Biochemistry, University of Toronto, Toronto, ON M5G 0A4, Canada; 8Department of Paediatrics, University of Toronto, Toronto, ON M5G 0A4, Canada

**Keywords:** cystic fibrosis, high throughput, in vitro models, CFTR, ENaC, ion channel activity

## Abstract

Induced Pluripotent Stem Cells (iPSCs) can be differentiated into epithelial organoids that recapitulate the relevant context for CFTR and enable testing of therapies targeting Cystic Fibrosis (CF)-causing mutant proteins. However, to date, CF-iPSC-derived organoids have only been used to study pharmacological modulation of mutant CFTR channel activity and not the activity of other disease-relevant membrane protein constituents. In the current work, we describe a high-throughput, fluorescence-based assay of CFTR channel activity in iPSC-derived intestinal organoids and describe how this method can be adapted to study other apical membrane proteins. Specifically, we show how this assay can be employed to study CFTR and ENaC channels and an electrogenic acid transporter in the same iPSC-derived intestinal tissue. This phenotypic platform promises to expand CF therapy discovery to include strategies that target multiple determinants of epithelial fluid transport.

## 1. Introduction

There has been remarkable progress made in the use of patient tissue-derived primary organoids for the in vitro modeling of Cystic Fibrosis (CF) pathogenesis and testing of therapies targeting mutant Cystic Fibrosis Transmembrane Conductance Regulator (CFTR) [1,2,3,4,5]. CFTR mutations lead to the loss of CFTR expression and/or function as a cyclic AMP-regulated anion channel at the cell surface. In three-dimensional (3D) primary organoids, the addition of forskolin results in higher intracellular cAMP, which activates CFTR channel activity and leads to swelling of the luminal cavity [2]. CFTR mutations that are associated with CF reduce forskolin-dependent swelling while also enabling the ranking of therapeutic interventions targeting defective CFTR expression and function [2,6]. Organoid swelling has been shown to correlate with multiple clinical biomarkers of CF, such as high sweat chloride concentration and compromised lung function measured as FEV_1_ [6]. This demonstrates the relevance of patient-derived organoids for in vitro assessment of patient-specific responses to modulators that directly target mutant CFTR.

However, such 3D structures have not been useful for screening the activities of cation channels and electrogenic transporters implicated in net epithelial fluid absorption. Previously, 2D monolayer cultures were generated from enzymatically dissociated rectal organoids to provide direct access to the apical membrane [7]. However, while these 2D cultures enabled low-throughput electrophysiological assays of CFTR-mediated chloride conductance, they did not reconstitute the native functional expression of ENaC, even though this channel is known to be expressed in the large intestine [8]. 

Merkert et al. developed 2D intestinal epithelial cultures from CF-Induced Pluripotent Stem Cells (iPSC) and demonstrated the potential of this model for high-throughput drug screening of novel CF therapies [5]. In this case, a halide sensitive reporter protein (eYFP) was genetically integrated, enabling studies of anion channel activity. As of yet, no similar strategy has been developed for the study of cation channels in iPSCs [9].

We reasoned that enzymatic dissociation of the 3D organoids as in the above study is too disruptive [7] and that by allowing the organoids to *open* to form a 2D lawn, we would preserve apical expression of physiologically relevant, sodium-dependent channels and transporters. In the current work, we show that the “opening” procedure does, in fact, retain 3D expression levels of ENaC and is adaptable to medium–high-throughput, high-content, phenotypic analyses.

Given the importance of ENaC and sodium-dependent transporters for modifying net epithelial fluid transport and maintaining epithelial barrier function, there is a clear need for the development of robust models and assays to test these activities and their modulation. It has been proposed that they could serve as potential molecular targets for companion therapies to augment the impact of approved CFTR modulators drugs [9,10,11]. Further, with the adaptation of these methods to the study of patient-specific tissues, we augment the diagnostic toolkit required for precision medicine. 

## 2. Materials and Methods

### 2.1. ESC (Embryonic Stem Cell) and iPSC-Derived Human Intestinal Organoids (HIO)

Table 1 defines the ESC and iPSC lines employed for organoid differentiation. H1-derive ESC HIOs were provided by Cincinnati Children’s Hospital Pluripotent Stem Cell Facility (PSCF). iPSCs were provided by the Cystic Fibrosis Individualized Therapy (CFIT) Program [12]. CRISPR-edited isogenic controls for the iPSC lines (non-CF1 and non-CF2) were generated by the Centre for the Commercialization of Regenerative Medicine as previously described [12]. Briefly, CRISPR-Cas9 with CFTR-targeted gRNA and Wt ssODN repair was used to generate correction of the homozygous mutation (F508del). 

Human intestinal organoids were differentiated as previously described [13]. In brief, iPSCs were cultured on Matrigel^®^ (Corning Inc., NewYork, NY, USA, CACB356231) coated plates in mTeSR1 media (STEMCELL Technologies, Vancouver, BC, Canada, 5850). At approximately 60% confluency, differentiation to definitive endoderm was initiated through the addition of Activin-A (100 ug/uL, R&D Systems, Minneapolis, MI, USA, 338-AC-050) for 3 days. Cultures were then exposed to hindgut endoderm differentiation media containing FGF4 (500 ng/mL, R&D Systems, 235-F4-025) and Chiron99021 (3 µM, R&D Systems, 4423/10) for 4 days. Post-hindgut differentiation, budding immature organoids were collected and embedded into 50 µL solid Matrigel^®^ (Corning Inc., CACB356231) drops. Spheroids were cultured for 30 days in previously established growth factor conditioned media [14].

After maturation, HIOs were passaged every 7–10 days and media was changed once every 2 days. To passage organoids, organoids were first collected in ice-cold PBS (Wisent, St. Bruno, QC, Canada, 311-010-CL) and pelleted through centrifugation for 5 min at 300× *g* at 4 °C. Post-centrifugation, excess PBS (Wisent, 311-010-CL), Matrigel^®^ (Corning Inc., CACB356231) and cellular debris was aspirated. The pelleted organoids were re-suspended in 1 mL of GCDR (STEMCELL Technologies, 7174) and incubated at room temperature for 5 min. With a P1000 pipettor, the organoids were fragmented through pipetting 40–60 times. Organoid fragments were then pelleted through centrifugation and re-suspended in fresh Matrigel^®^ (Corning Inc., CACB356231) domes and seeded at a 1:3 ratio. Growth factor conditioned medium was added to after allowing the Matrigel^®^ (Corning Inc., CACB356231) to solidify at 37 °C for 35 min. Detailed descriptions of HIOs are listed in Table 1.

### 2.2. MDCK and MDCK (Tagged αβγENaC) Cells

As previously described [15], both MDCK and MDCK (HA-tagged αENaC, myc-tagged βENaC, FLAG-tagged γENaC) cells are grown in DMEM media (Invitrogen, Waltham, MA, USA, 12634010), supplemented with 10% FBS (Wisent, 320-005-CL), 1% Penicillin/Streptomycin solution (Wisent, 450-200-EL) and additional selection antibiotics (300µg/mL G418 (Wisent, 400-130-IG), 100µg/mL Hygromycin B (Thermofisher, Waltham, MA, USA, 10687010) and 2µg/mL Puromycin (Thermofisher, A1113802)). MDCK cells are cultured in the presence of Amiloride (10 µM, Spectrum Chemical, New Brunswick, NJ, USA, TCI-A2599-5G). Twenty-four hours prior to ASC assay, αENaC expression in MDCK cells was induced with dexamethasone (1 µM, Sigma-Aldrich, St. Louis, MI, USA, D4902) and sodium butyrate (10µM, Sigma-Aldrich, B5887) to enhance protein expression of ENaC subunits.

### 2.3. Swelling Assay

iPSC-derived organoids were isolated from the Matrigel^®^ support using ice-cold HBSS (Hank’s Balanced Salt Solution, Wisent, 311-513-CL) and pelleted through centrifugation at 500 g for 5 min. The cell pellet was re-suspended in HBSS containing 3 µM of live cell maker dye, Calcein-AM (Sigma-Aldrich, St. Louis, MO, USA, 17783) at 37 °C for 45 min. Excess dye was removed through centrifugation, and organoids were re-suspended in DMEM/F12 (Invitrogen, 12634010) for human intestinal organoids. Organoid swelling was induced using Fsk (Sigma-Aldrich, F3917) for non-CF organoids or Fsk in combination with CFTR potentiators. Human intestinal organoid swelling was performed in 96-well plates, with each condition containing at least 50 organoids per biological replicate. The swelling was tracked for 4 h and imaged at 15 min intervals using confocal microscopy (Nikon A1R Confocal Laser Microscope). 

Organoid swelling analysis was performed using Cell Profiler v3.1 (Cambridge, MA, USA). Swelling images were analyzed using an in-house developed algorithm, as previously described [10]. In brief, images were exported as individual TIFF files and aligned using translation registration by cross-correlation. A histogram-derived thresholding method (triangle) was used to identify specific organoids in the images. The center of the object mask was used to track individual organoids throughout the experiment. The differential organoid size at each time point was calculated by subtracting the size of the initial timepoint. Organoids were excluded if they were not tracked for the entire time course.

### 2.4. Opened Organoid Cultures

Organoids were removed from the Matrigel^®^ (Corning Inc., Corning, NY, USA, CACB356231) domes and collected in ice-cold DMEM/F12 (Invitrogen, 12634010) and pelleted through centrifugation at 500× *g* for 5 min. Pellets were then re-suspended in growth factor conditioned medium and plated onto Poly-L-Lysine (0.01% solution, Sigma-Aldrich, P4707) coated plates (Corning Inc., 3916). Organoids were collected and suspended at a density of approximately 1000 organoids per 1 mL of media. This suspension was used to seed wells in a 96-well plate such that after opening, 50% of the surface area is covered (Appendix A). Media were changed one-day post-seeding. All functional studies were conducted two days after organoid plating. With the apical application of specific electrochemical gradients, ligands and inhibitors, we selectively measure the apical electrogenic membrane protein function.

### 2.5. Apical Chloride Conductance (ACC) Assay for CFTR Function

The ACC assay was used to assess CFTR-mediated changes in membrane depolarization using methods previously described [16]. In summary, opened iPSC-derived intestinal organoids were incubated with zero sodium, chloride and bicarbonate buffer (150 mM NMDG (Sigma-Aldrich, M2004), 150 mM Gluconic Acid Lactone (Sigma-Aldrich, G4750), 3 mM Potassium Gluconate (Sigma-Aldrich, P1847), 10 mM Hepes (Bioshop, Burlington, ON, Canada HEP001.5), pH 7.42, 300 mOsm) containing 0.5 mg/mL of FLIPR^®^ dye (Molecular Devices, San Jose, CA, USA R8042) for 30 min at 37 °C. The Wt-CFTR function in iPSC-derived gene-edited organoids was measured after the acute addition of Fsk (10 µM, Sigma-Aldrich, F3917) or DMSO (0.1%, Sigma-Aldrich, M81802) control. In iPSC-derived F508del CF organoids, cells were chronically rescued with corrector compounds for 24 h using 3 µM VX-809 (Selleck Chemicals, Houston, TX, USA, S1565), 3 µM VX-445 (MedChemExpress, Monmouth Junction, NJ, USA, HY-111772) / 3 µM VX-661 (Selleck Chemicals, S7059) or DMSO control. Post-CF corrector modulator rescue, the F508del-CFTR function was measured after the acute addition of 10 µM Fsk (Sigma-Aldrich, F3917) and 1 µM VX-770 (Selleck Chemicals, S1144). Additional modulators (8cGMP (Sigma-Aldrich, 15992), GSNO (Cayman Chemicals, Ann Arbor, MI, USA, 82240), Milrinone (Sigma-Aldrich, M4659) and Tadalafil (Sigma-Aldrich, SML1877)) were added during the FLIPR^®^ dye-loading process. CFTR functional recordings were measured using the FLIPR^®^ Tetra High-throughput Cellular Screening System (Molecular Devices), which allowed for simultaneous image acquisition of the entire 96-well plate. Images were first collected to establish baseline readings over 5 min at 30 s intervals. Modulators were then added to stimulate CFTR-mediated anion efflux. Post-drug addition, CFTR-mediated fluorescence changes were monitored, and images were collected at 15 s intervals for 70 frames. CFTR channel activity was terminated with the addition of 10 µM CFTRInh172 (CF Foundation Therapeutics), and fluorescence changes were monitored at 30 s intervals for 25 frames.

### 2.6. Apical Sodium Conductance (ASC) Assay for ENaC Function

The ASC assay was used to assess ENaC inhibition upon amiloride addition through assessing changes in membrane hyperpolarization. Opened iPS-derived intestinal organoids were incubated with a physiological sodium gluconate buffer (150 mM Sodium Gluconate (Sigma-Aldrich, G9005), 3 mM Potassium Gluconate (Sigma-Aldrich, P1847), 10 mM Hepes (Bioshop, HEP001.5), pH 7.42, 300 mOsm), containing FLIPR^®^ dye (0.5 mg/mL), for 30 min at 37 °C [17]. After dye loading, the plate was transferred to the FLIPR^®^ Tetra High-throughput Cellular Screening System (Molecular Devices). Baseline readings were acquired for 5 min at 30 s intervals. In *opened* organoids, ENaC inhibition was measured following the acute addition of Amiloride (10 µM or 50 µM, Spectrum Chemical, TCI-A2599-5G), Phenamil (10 µM or 50 µM, Sigma-Aldrich, P203) or DMSO control. Inhibitor-mediated membrane hyperpolarization was tracked over time as a loss in fluorescence signal over 70 min at 60 s intervals.

### 2.7. Apical Amino Acid Conductance (AAC) Assay for SLC6A14 Function

The AAC assay was used to measure SLC6A14-mediated acute uptake of Arginine leading to membrane depolarization. iPSC-derived gene-edited organoids and F508del CF organoids (rescued chronically for 24 h with VX-809 or DMSO control) were incubated in a low-sodium, low-chloride buffer (112.5 mM NMDG (Sigma-Aldrich, M2004), 112.5 mM Gluconic Acid Lactone (Sigma-Aldrich, G4750), 36.25 mM NaCl (Sigma-Aldrich, S9888), 2.25 mM Potassium Gluconate (Sigma-Aldrich, P1847), 0.75 mM KCl (Sigma-Aldrich, P3911), 0.75 mM CaCl_2_ (Sigma-Aldrich, C1016), 0.5 mM MgCl_2_ (Sigma-Aldrich, M8266) and 10 mM HEPES (Bioshop, HEP001.5), pH 7.42, 300 mOsm), containing FLIPR^®^ dye (0.5 mg/mL) for 40 min at 37 °C [18]. During dye loading, organoids were treated with α-MT (2 mM, Sigma-Aldrich, M8377) or buffer control. After dye loading, the plate was transferred to the FLIPR^®^ Tetra High-throughput Cellular Screening System (Molecular Devices). Baseline readings were acquired for 5 min at 30 s intervals. Arginine (1 mM) was added acutely to opened organoids pretreated with SLC6A14 blocker, α-MT [19,20]. Change in fluorescence was recorded at 30 s intervals.

### 2.8. FLIPR Analysis and Heatmap Generation

Experiments were exported as multi-frame TIFF images, for which every frame is recorded in the entire plate. Pixels outside of the well areas were filtered out using the initial signal intensities, and wells containing *opened* organoids were separated. All traces were normalized to the last point of the baseline intensity. Peak response for each pixel was calculated as the maximum deviation from baseline. During the stimulation segment, the fluorescence intensity increased for CFTR and decreased for ENaC function. A heatmap representation was generated from the peak response of each pixel, and the mean response trace of wells was generated by averaging the corresponding pixel traces. The CFTR and ENaC-dependent FLIPR signals described are independent of cell number, as we are reporting the average of peak responses detected across an entire scan. The resolution of the image enables measurement from 2–3 cells per pixel. The analysis methods were previously described in detail [16].

### 2.9. Real-Time Quantitative PCR

As previously described [21], 4 wells of 96-well plate at 40–50% confluency of *opened* organoid samples were collected per sample in ice-cold Phosphate Buffered Saline and pelleted through centrifugation at 500 g for 5 min. Total mRNA from pelleted samples was extracted using RNeasy^®^ Plus Micro Kit (Qiagen, 74004), following the enclosed instructions. After confirming the spectrophotometric quality of extracted RNA through 260/280 ratios of 2.0 and 260/230 ratios of 1.8–2.2, mRNA samples with concentrations greater than 300 ng/µL were used to reverse transcribe 1 µg of cDNA using the iScript^TM^ cDNA Synthesis Kit (BioRad, 170-8891). Expression levels of target genes were measured using primers listed in Table 2 using the FX96 Touch^TM^ Real-Time PCR Detection System using the SYBR Green Master Mix containing the EvaGreen^®^ Fluorophore with low ROX (BioRad, 58343). The expression of the target genes was normalized to housekeeping gene control TBP (TATA Box binding Protein) [10]. Primer sequences for gene expression studies are listed in Table 2.

### 2.10. Immunofluorescence

Four wells of *opened* organoids were collected from 96-well plates. Organoids were fixed and permeated with 100% methanol at −20° C for 10 min. Post-methanol incubation, samples were washed 3 times with PBS, 5 min per wash, at room temperature. Following the washes, samples were blocked using 4% BSA for 30 min and incubated with primary antibodies against CDX2 (1:100, Abcam, ab76541) or Zona Occluden-1 (ZO-1) (1:200, Thermofisher, 61-7300) and CFTR overnight. After the removal of the primary antibody, samples were washed 3 times with PBS, 5 min per wash, and incubated with secondary antibodies (Alexa Fluor 594 Monoclonal (Thermofisher, R37119) and Alexa Fluor 488 Polyclonal Antibody (Thermofisher, R37114)) and nuclear marker DAPI (1:400, Thermofisher, D1306) for 1 h. Samples were then washed 3 times with PBS, 5 min per wash, at room temperature. Confocal images were taken using Nikon A1R Confocal Laser Microscope.

### 2.11. Western Blotting

Samples were collected in ice-cold PBS and pelleted through centrifugation at 4 °C (500× *g* for 7 min). Post-centrifugation, the cell pellet was re-suspended in 200 μL of modified radioimmunoprecipitation assay buffer (50 mM Tris-HCl, 150 mM NaCl, 1 mM EDTA, pH 7.4, 0.2% (*v/v*) SDS and 0.1% (*v/v*) Triton X-100) containing a protease inhibitor cocktail for 10 min. After centrifugation at 13,000 rpm for 5 min, the soluble fractions were analyzed by SDS-PAGE on 6% Tris-Glycine gel. After electrophoresis, proteins were transferred to nitrocellulose membranes and incubated in 5% milk, and CFTR bands were detected using the mouse monoclonal antibody 596 (1:1000, UNC CFTR antibodies). CFTR expression is measured relative to the internal loading control protein, calnexin. Calnexin (CNX, Sigma-Aldrich, C4731) was detected using a Calnexin-specific rabbit polyclonal antibody (1:5000). MDCK and MDCK (tagged αβγENaC) cell lysates were αENaC, βENaC, and γENaC subunits were detected using antibodies against the epitope tags, anti-HA (BioLegend, San Diego, CA, USA, 901503), anti-Myc (Millipore, Burlington, MA, USA, 05-724), and anti-Flag antibodies (Cell Signaling, Danvers, MA, USA, 14793S). In MDCK and MDCK (tagged αβγENaC) cell, αENaC, βENaC, and γENaC expressions are measured relative to internal loading control protein, Actin (Sigma-Aldrich, A228). After primary antibody incubation, blots were washed 3 times with PBS and incubated in secondary antibodies Anti-rabbit IgG (Abcam, ab6721) and Anti-mouse IgG (Abcam, ab6789) (1:5000). The blots were developed using the Li-Cor Odyssey Fc (LI-COR Biosciences, Lincoln, NE, USA) in a linear range of exposure (1–20 min).

### 2.12. Statical Analysis

One-way ANOVA with Tukey’s multiple comparison was performed on data with more than two datasets. Unpaired t-test was performed on data with two datasets. Statistical analysis was performed using Graphpad Prism version 9.2.0 (San Diego, CA, USA). *p* < 0.05 was considered to be significant.

## 3. Results

### 3.1. 3D CF Human Intestinal Organoids (HIO) Exhibit Defective Fluid Secretion, Which Can Be Restored through the Use of CFTR Modulators or Gene Editing

HIOs were differentiated from Embryonic Stem Cells (H1 ESC line) and iPSC lines (non-CF1 and non-CF2 (*n* = 2) and CF1 (*n* = 1), see Table 1) using previously established differentiation protocols (Figure 1a) [13,14]. Immunostaining confirmed that the CF1 and non-CF1 HIOs expressed CDX2, and Zona Occludin-1 (ZO-1), which are characteristic of intestinal epithelial cells (Figure 1b). A similar spheroid morphology was detected regardless of the CFTR genotype, in contrast to previous studies comparing primary CF and non-CF rectal organoids (2). We expected a smaller luminal cavity size in CF HIOs. The appearance of a lumen under baseline, unstimulated conditions in CF organoids suggests the contribution of non-CFTR channels that remain to be identified. 

However, a primary functional difference was revealed by comparing Fsk-induced swelling. HIOs differentiated from mutation corrected non-CF1 iPSCs, which harbors Wt-CFTR, exhibited an increase in size after activation by the adenylate cyclase agonist, Forskolin (Fsk), unlike the organoids differentiated from CF iPSCs (CF1). These findings are expected and confirm that CF intestinal organoids lack CFTR-mediated forskolin-induced swelling (Figure 1c,d) [2].

In CF HIOs, a defective F508del-CFTR function (impaired Fsk-induced swelling) was rescued by treatment with the CFTR corrector, lumacaftor (VX-809) for 24 h, followed by acute potentiation with ivacaftor (VX-770) (Figure 1c,d), as expected from previous studies [14].

### 3.2. Opened HIOs Enable Direct Assessment of Apical Wt-CFTR Channel Function in a High-Throughput Format

Electrophysiological assays of CFTR modulator activity in 2D intestinal monolayer cultures are considered the “gold standard” for testing efficacy. However, these methods are relatively low throughput. Thus, we were prompted to develop a complementary 2D model and assay of CFTR function that would enable direct measurement of F508del-CFTR channel modulation at the apical membrane of the HIOs.

As previously demonstrated using primary mouse colonic organoids [10], the removal of the supporting extracellular matrix led to the splitting open of organoids and direct access to the apical membrane. We applied this method to the study of iPSC-derived HIOs to generate 2D *opened* organoids (Figure 2a). Characterization studies (RT-qPCR) confirmed there was no significant reduction in the expression of *ENaC*, and the electrogenic amino acid transporter, *SLC6A14*, and other intestinal and epithelial cell markers, when comparing 2D *opened* organoids to 3D HIOs (Figure 2b, Appendix A). In addition, we show that mature CFTR protein (band C) is expressed in 3D HIOs and *opened* HIOs, albeit the abundance is reduced in the *opened* structure in this study (Figure 2c).

Immunofluorescence studies were conducted to confirm apical membrane location through visualization of tight junction complex protein, Zona Occluden-1 (ZO-1) and CFTR expression in the *opened* non-CF HIOs (Figure 2d). Interestingly, there appeared to be discrete cell types that express relatively high apical expression of CFTR. Single-cell analytical methods will be employed to identify the cells with high CFTR signals in the future. 

After the confirmation of CFTR protein expression, CFTR channel function was measured in *opened* HIOs using the Apical Chloride Conductance (ACC) assay (Figure 3a), as previously shown in mouse colonic organoids [10]. From 3D H1 HIOs (Table 1), we subsequently generated *opened* organoids to test CFTR-mediated changes in membrane potential following Fsk stimulation. After generating H1 *opened* organoids, cultures treated with Fsk displayed an increase in fluorescence. CFTR stimulation was measured as an average fluorescence increase across the entire well. CFTR function was inhibited with addition of CFTRInh-172 (Figure 3b,c). We then measured CFTR responses in the non-CF *opened* HIOs, expressing Wt-CFTR (Table 1, Figure 3d). The *opened* non-CF HIOs showed excellent reproducibility of peak response stimulation and consistent activation kinetics with Fsk stimulation and inhibition with CFTRInh-172, reporting a Z’ factor of 0.5294, supporting its utility as a robust assay of dynamic CFTR function in a high-throughput format (Figure 3e, Appendix A). To demonstrate reproducibility across different iPSC lines, we differentiated and generated *opened* organoids from another non-CF line (non-CF2) to assess CFTR channel activity using the ACC assay. Consistent with the non-CF1 *opened* organoids, CFTR stimulation in the non-CF2 *opened* organoids was detected with the addition of Fsk relative to the DMSO controls (Figure 3f,g). Finally, we show the Fsk dose-response of the ACC assay with an EC50 of 0.0287 μM (Figure 3h). 

### 3.3. Opened CF Organoids Can Model Pharmacological Rescue of F508del-CFTR with CF Modulators

We were prompted to determine if the ACC assay is effective in both detecting the primary defect caused by the F508del mutation and evaluating the efficacy of clinical modulators on *opened* CF HIOs (Figure 4a). The *opened* F508del CF1 HIOs displayed no significant fluorescence changes with Fsk stimulation without modulator rescue, which is consistent with the expected defect in F508del-CFTR channel function prior to modulator rescue (Figure 4b). VX-809 (3 µM)/VX-770 (1 µM) treatment resulted in significant partial rescue of the mutant F508del-CFTR protein [23]. Further treatment with the new and highly effective modulator combination, VX-661 (3 µM), VX-445 (3 µM) and VX-770 (1 µM) (TRIKAFTA^TM^) [24], restored F508del-CFTR function to approximately 50% of Wt-CFTR function of non-CF HIOs (Figure 4b–d). 

To determine if iPSC HIOs have the potential to identify companion therapies for CF, we assessed the effects of known modulators of PKA and PKG phosphorylation since the phosphorylation of CFTR by PKA and PKG kinases have been implicated in regulating modulator efficacy (Figure 4e) [25,26]. After partial correction of the F508del trafficking defect using VX-809, *opened* F508del CF1 HIOs were acutely potentiated with Fsk/VX-770 in combination with a nitric oxide (NO) agonist, which targets the enhancement of the PKG phosphorylation pathway (8cGMP or GSNO) and has been reported to augment VX-809 rescued F508del-CFTR activity [16,18]. Alternatively, *opened* F508del CF1 HIOs were treated with phosphodiesterase inhibitors (Milrinone or Tadalafil), which have been shown to be effective in the stimulation of F508del-CFTR short circuit currents in murine intestinal tissue [26]. Milrinone addition, along with VX-809/VX-770, significantly increased the Fsk response to levels comparable to the triple modulator combination (VX-661/VX-445/VX-770) treatment (Figure 4e,f). Therefore, the ACC assay is sufficiently sensitive and precise to distinguish between various modulator combinations that are expected to exhibit different efficacies in rescuing the functional expression of F508del-CFTR. Direct access to the apical membrane of HIOs in the *opened* format prompted us to determine whether the functional output of other apical membrane channels can be detected.

### 3.4. Measurement of ENaC Specific Activity in MDCK Overexpression Cells

With access to the apical membrane, we now have the opportunity to study other apical membrane channels and transporters that have been implicated in CF pathophysiology [9,11,27,28,29]. Since ENaC is functionally expressed in the intestinal epithelium [30] but has not been studied in a high-throughput manner in patient-derived primary intestinal organoids [7], we were prompted to develop a novel assay ENaC measuring mediated changes of membrane potential in opened organoids.

As a proof-of-concept, we first determined if the FLIPR^®^ membrane potential dye is capable of measuring ENaC function in a well-characterized cell line either lacking ENaC expression or overexpressing ENaC. This involved the use of the renal epithelial MDCK cell line, genetically engineered to express an HA-tagged αENaC, a myc (and T7)-tagged βENaC and a FLAG-tagged γENaC and its non-transfected parental MDCK control line (Figure 5a) [27]. In order to measure the constitutive ENaC function, we modified the existing ACC assay measuring the CFTR function and established an inward driving sodium gradient to assess the effect of ENaC inhibitors, amiloride and phenamil, on the apical membrane of confluent, differentiated MDCK monolayers. Under these conditions, we predicted that the apical membrane potential, as monitored by the novel Apical Sodium Conductance (ASC) assay, would hyperpolarize upon inhibition of ENaC with amiloride or phenamil (Figure 5b).

As expected, MDCK cells expressing tagged αβγENaC also exhibited membrane hyperpolarization after the addition of either amiloride or phenamil at both 10 and 50 µM concentrations only in the presence of extracellular sodium (Figure 5c,d). In the absence of extracellular sodium, the addition of amiloride 10 or 50 µM does not result in membrane hyperpolarization (Appendix A), which demonstrates that the membrane hyperpolarization effect with the addition of amiloride is sodium-dependent. These responses were significantly greater in the MDCK cells expressing tagged αβγENaC relative to the parental line. At 10 µM concentrations of these inhibitors, with phenamil known to be highly specific for ENaC, there was a clear and significant difference between the transfected and un-transfected MDCK lines. These findings support the claim that this ASC assay captures ENaC activity. At the higher concentration of these inhibitors of 50 µM, there was still a significant difference between transfected and untransfected cells. However, there is a possibility that 50 uM of these inhibitors could inhibit NHE3 [31], and through changes in intracellular pH, indirectly affect other channels.

### 3.5. Function of ENaC Activity Can Be Measured in Both Opened CF HIOs and Non-CF HIOs and HCOs

Once we confirmed that the FLIPR assay could detect ENaC function specifically, we asked if the ENaC function was measurable in *opened* HIOs using the ASC (FLIPR) assay described above. In fact, as in the transfected MDCK cell line, the acute addition of phenamil (10 µM) or amiloride (10 µM) did evoke hyperpolarization (Figure 6a,b) [32]. Similar to MDCK cells expressing tagged ENaC, the addition of amiloride at a higher concentration (50 µM) resulted in a greater reduction in ENaC function, although the response measured as 50 µM may reflect both ENaC activity as well as indirect effects of inhibiting the sodium-proton exchanger, NHE3. No difference was detected in pilot studies comparing the magnitude of amiloride (50 µM)-sensitive responses in opened CF intestinal organoids (CF1) and organoids differentiated from the isogenic control (non-CF1).

Interestingly, the absolute magnitude of the phenamil-mediated (10 µM) FLIPR response was similar between the ENaC subunit-transfected MDCK cultures and the opened HIOs organoid cultures that are expected to express much less of the sodium channel transcript and protein. With the future identification of a specific ENaC antibody that detects endogenous ENaC protein, we will be able to test the hypothesis that the FLIPR assay detects ENaC localized to the cell surface and that this is comparable in the two systems.

Similar to the ACC assay of the CFTR channel function (Figure 2d), the ASC assay of ENaC function in *opened* iPSC-derived HIOs showed excellent reproducibility and a consistent amiloride response, resulting in a Z’ factor of 0.573. Such parameters indicate the *opened* model and ASC assay together provide an excellent candidate platform for monitoring dynamic and high-throughput drug screening of potential ENaC modulators (Appendix A), further validating the suitability of the ASC *opened* HIOs for drug screening and the evaluation of modulator efficacy (Appendix A).

### 3.6. Specific Activity of Sodium and Chloride-Dependent Amino Acid Transporter, SLC6A14, Can Be Measured in Opened CF and Non-CF HIOs

Previously, the *opened* organoid model was applied to mouse colonic organoids in order to determine the activity of the sodium-dependent amino acid transporter, SLC6A14 [10]. SLC6A14 is a sodium- and chloride-dependent electrogenic amino acid transporter expressed in airway, intestinal and colonic epithelial tissues. SLC6A14 mediates the uptake of cationic and neutral amino acids, along with two sodium ions and one chloride ion, generating one net positive charge translocation and membrane depolarization per amino acid transport [10,19]. Through the application of a low-sodium/chloride extracellular gradient, SLC6A14-mediated depolarization could also be measured in *opened* HIOs (Figure 7a). In the presence of low extracellular sodium and chloride, the addition of arginine (L-Arg) to the apical surface of these *opened* organoids evokes apical membrane depolarization in both *opened* non-CF and CF HIOs. Based on previous mechanistic studies [20], the pretreatment using the SLC6A14 blocker, α-Methyl-DL-tryptophan (α-MT) should block SLC6A14-mediated amino acid uptake [10,20]. As expected, this signal was abolished when pretreated with α-MT (Figure 7a–c) [10,19]. Stimulation with acute L-Arg treatment and blockage of this stimulation using α-MT suggests that functional measurements of SLC6A14 amino acid uptake can be achieved using *opened* HIOs. Hence, there is the potential to identify SLC6A14 modulators in this platform for use in both CF and non-CF intestinal tissue.

## 4. Discussion

In the current work, we described novel methods for studying sodium-dependent, apical membrane ion channel and transporter function in intestinal organoids differentiated from ESCs and iPSCs. By “opening” organoids to expose the apical membrane, we enabled the measurement of CFTR, ENaC and the electrogenic amino acid transporter, SLC6A14. Therefore, multiple CF disease-relevant, electrogenic membrane proteins can be interrogated in the same patient-derived tissue [8,9,10]. With these innovations, we have expanded the potential application of CF organoid models to include testing on multiple therapeutic targets.

The Apical Chloride Conductance (ACC) and Apical Sodium Conductance (ASC) assays of patient-derived tissues exhibit excellent reproducibility with Z’ factor scores of 0.529 and 0.573, respectively. Because of the capacity of this platform for profiling multiple small molecule combinations simultaneously, we found that phosphodiesterase inhibitors could be used as a companion therapy, in combination with ORKAMBI^TM^, to boost F508del-CFTR chloride channel activity to levels comparable to that achieved by the new triple modulator combination, TRIKAFTA^TM^. Therefore, the *opened* organoid model can serve as a tool in identifying alternative therapeutics for patients with limited access to TRIKAFTA^TM^. Further, the *opened* organoids have the potential for identifying proposed ENaC modulators in a high-throughput format, which can be further investigated and characterized in electrophysiological studies using the Ussing chamber.

The main purpose of the current study was to develop methods using the FLIPR membrane potential dye for quantifying multiple apical channels and transporters in non-CF intestinal tissue differentiated from embryonic cells and iPSCs. Toward this goal we studied three different non-CF lines. However, we did conduct pilot studies, comparing ENaC activity in *opened* intestinal organoids differentiated from the CF iPSC cell (CF1, harbouring F508del) and its isogenic, non-CF control. Interestingly, we did not observe a significant difference in the ENaC activity between the CF and non-CF samples. This was somewhat surprising given previous findings that ENaC function is augmented in CF tissues [27]. However, we plan to repeat these studies using the multiple CF lines with matching isogenic non-CF lines in the CFIT resource [12] to determine if there may be differences between donors.

Our previous studies showed that arginine transport through the SLC6A14 amino acid transporter indirectly augments Wt-CFTR and F508del-CFTR channel activity in primary mouse colonic organoids via cGMP and nitric oxide-mediated signaling [10]. Here, we show that SLC6A14 is functionally expressed in the apical membrane of CF and non-CF intestinal organoids differentiated from iPSC. Interestingly, we did not observe a similar augmentation of F508del-CFTR activity in human intestinal organoids by agonists of nitric oxide (Figure 4f). For the reason discussed above, future studies are required to fully understand the role of SLC6A14 in the modification of human intestinal disease in CF patients.

In summary, we demonstrated the ability to detect the function of multiple apical membrane proteins in human tissues in a format suitable for in-depth analysis of ion channel regulation and interaction. The high-content and high-throughput capacity of this format will be suitable for the identification of novel modulators of mutant CFTR as well as other modulators of other channels and transporters implicated in CF pathogenesis.

## Figures and Tables

**Figure 1 cells-10-03419-f001:**
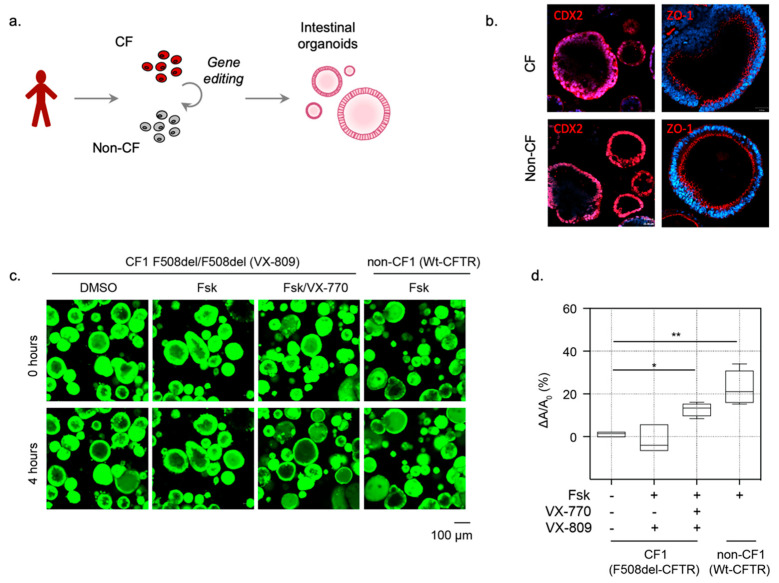
iPSC-derived HIOs can be used to measure the function of Wt-CFTR and pharmacologically rescued F508del-CFTR as Fsk-induced organoid swelling. (**a**) Schematic depicting the generation of 3D Human Intestinal Organoids (HIOs) from patient-derive iPSCs. (**b**) Characterization of iPSC-derived CF and non-CF intestinal organoids. Immunofluorescence studies of iPSC-differentiated HIOs highlighting expression of intestinal cell markers CDX2 (red, intestinal marker), and ZO-1 (red, epithelial cell/tight junction marker). (**c**) Representative images of Fsk-induced swelling of F508del-CFTR expressing CF organoids and Wt-CFTR expressing (non-CF) organoids. CF (CF01) organoids were rescued chronically (24 h) with DMSO control or VX-809 (3 µM) and acutely stimulated with Fsk (10 µM) or Fsk and VX-770 (1 µM). (**d**) Box and whisker plot shows the change in organoid size post-Fsk-induced swelling (ΔA) relative to average organoid size at baseline (A_0_) (* *p* = 0.0191, ** *p* = 0.0065, *n* ≥ 3 biological replicates. Each biological replicate = independent organoid passage, technical replicate = average of >30 organoids). Box plot depicts the median value and bounds depict IQR ranges with the whiskers defining the minima and maxima values.

**Figure 2 cells-10-03419-f002:**
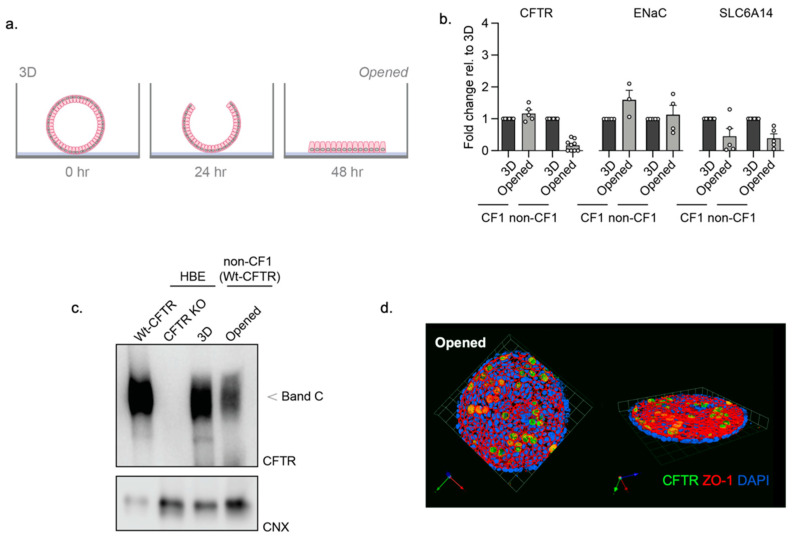
Characterization of *opened* and 3D iPSC-HIOs. (**a**) Schematic depicting the generation of *opened* organoids by removal of the extracellular supporting matrix. (**b**) Gene expression RT-qPCR studies of *CFTR* and *ENaC* and *SLC6A14* in *opened* organoids relative to 3D organoid expression. (**c**) Western blot of WT-CFTR expression in 3D and *opened* non-CF organoids, compared to expression in a Human Bronchial Epithelial (HBE) cell line. HBE cells expressing either WT-CFTR and or genetic knockout of CFTR as positive and negative controls for CFTR protein expression, respectively, as previously described [22]. (**d**) Immunofluorescence of CFTR (green), apical membrane marker, ZO-1 (red), and nuclei (blue) in *opened* non-CF01 organoids.

**Figure 3 cells-10-03419-f003:**
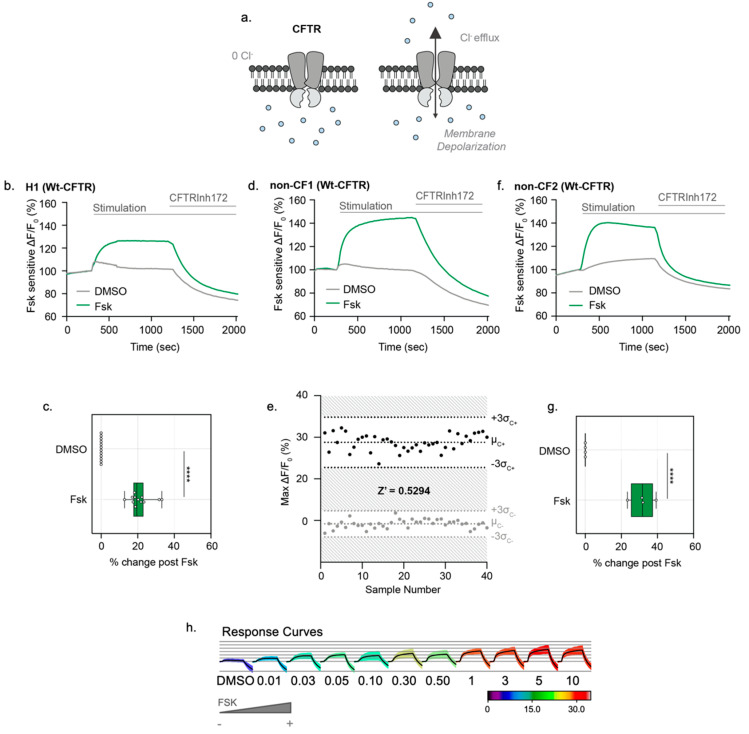
*Opened* iPSC-HIOs enable direct measurements of Wt-CFTR function in a high-throughput format. (**a**) Schematic depicting the Apical Chloride Conductance (ACC) assay. *Opened* iPSC-HIOs are placed in a zero-chloride extracellular buffer. Fsk stimulates CFTR-mediated chloride efflux, leading to an increase in membrane potential and an increase in fluorescence. The signal is terminated with acute treatment of CFTRInh172. (**b**) Representative trace and (**c**) Box and whisker plot of Wt-CFTR function measured in H1 Embryonic Stem Cell (ESC)-derive organoids treated with Fsk or DMSO control (**** *p* < 0.0001, *n* > 3 biological replicates, *n* > 3 technical replicates). Box plot depicts the median value and bounds depict IQR ranges with the whiskers defining the minima and maxima values. There are occasional minor deflections with DMSO addition, which we interpret as an artifact. This nonspecific response was subtracted during quantification shown in the bars and whiskers. (**d**) Representative trace of WT-CFTR function measured in *opened* non-CF organoids stimulated with Fsk (10 µM). CFTR response was terminated with CFTRinh172 (10 µM). (**e**) Bland–Altman plot depicting reproducibility of stimulated CFTR response. Black points measure the maximum change in fluorescence changes with acute Fsk stimulation in comparison to grey points representing DMSO control. (**f**) Representative traces and (**g**) box and whisker plot of Wt-CFTR function measured in non-CF2 iPSC-derived *opened* organoids treated with Fsk or DMSO control (**** *p* < 0.0001, *n* = 4 technical replicates). (**h**) Response curves of *opened* non-CF organoids stimulated with increasing concentrations of Fsk.

**Figure 4 cells-10-03419-f004:**
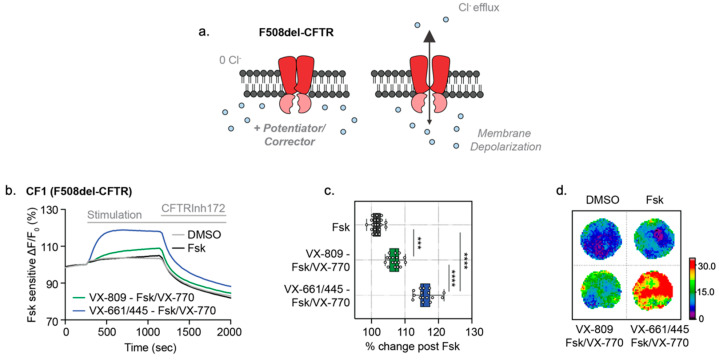
*Opened* CF HIOs can model defective CFTR function and response to CF modulators. (**a**) Schematic depicting functional measurement of F508del-CFTR and effect of CF modulators using the ACC assay. (**b**) Representative traces of F508del-CFTR response to modulator rescue in *opened* CF organoids. *Opened* F508del CF organoids were chronically (24 h) rescued with VX-809 (3 µM), VX-445/VX-661 (both 3 µM) or DMSO as control, and acutely stimulated with Fsk (10 µM), Fsk (10 µM)/VX-770 (1 µM) or DMSO control. (**c**) Box and whisker plot and (**d**) Peak response heatmaps of F508del-CFTR response to pharmacological rescue in *opened* CF organoids (*** *p* = 0.004, **** *p* < 0.001, *n* > 3 biological replicates, *n* = 3 technical replicates). (**e**) Table of cGMP, Nitric Oxide donors and Phosphodiesterase inhibitor compounds tested in combination with VX-770 and VX-809 to enhance F508del-CFTR modulator rescue. (**f**) Box and whisker plot shows F508del-CFTR peak response stimulation post-chronic rescue with VX-809 (3 µM) and acute drug treatment with test compounds and Fsk (10 µM)/VX-770 (1 µM) (**** *p* < 0.0001, *n* = 3 biological replicates, *n* = 3 technical replicates). Each biological replicate = independent organoid passage; technical replicate = 1 well of 96-well plate. Box plot depicts the median value and bounds depict IQR ranges with the whiskers defining the minima and maxima values.

**Figure 5 cells-10-03419-f005:**
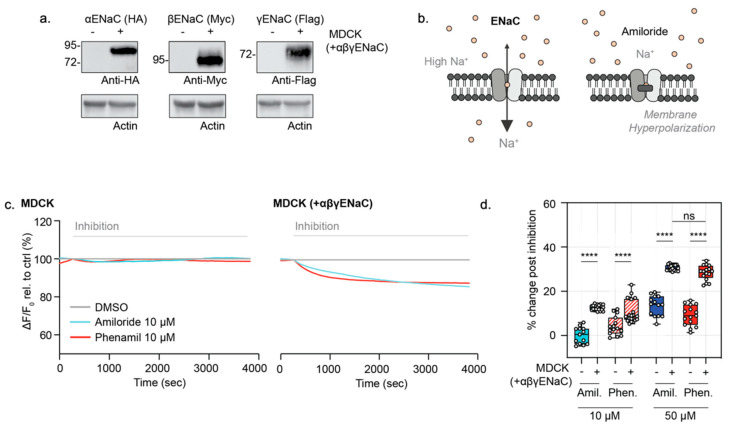
Validation of a novel ASC FLIPR assay of ENaC function in untransfected MDCK and αβγENaC transfected MDCK cells. (**a**) ENaC subunit expression in MDCK cells that are stably transfected compared to parental (non-transfected) MDCK control cells. Western blot detection of αENaC, βENaC or γENaC, with anti-HA, anti-Myc or anti-flag antibodies, respectively. (**b**) Schematic depicting ENaC inhibition in the novel Apical Sodium Conductance Assay (ASC). In the presence of the high extracellular sodium, acute addition of Amiloride or Amiloride analogues result in ENaC inhibition and relative membrane hyperpolarization, which is detected as a decreased fluorescence signal. (**c**) Representative traces of ENaC inhibition with amiloride and phenamil (10 µM) in MDCK parental cells and MDCK cells expressing tagged αβγENaC. (**d**) Box and whisker plot of ENaC inhibition in MDCK cells and MDCK cells overexpressing triple epitope-tagged αβγENaC with amiloride and amiloride analogue, Phenamil (10 µM and 50 µM) relative to DMSO control (**** *p* < 0.0001, *n* = 4 biological replicates, *n* ≥ 4 technical replicates). Each biological replicate = independent organoid passage; technical replicate = a well of 96-well plate. Box plot depicts the median value and bounds depict IQR (interquartile ranges) with the whiskers defining the minima and maxima values.

**Figure 6 cells-10-03419-f006:**
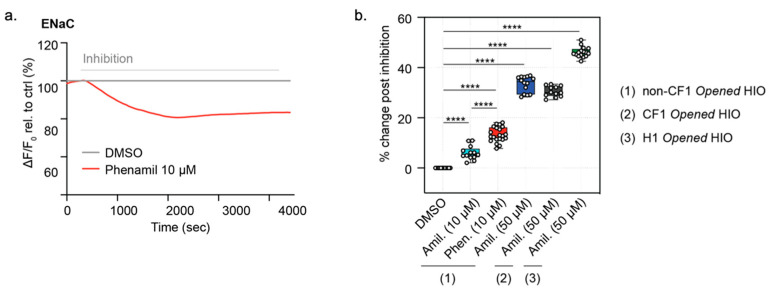
*Opened* iPSC-differentiated HIOs enable a high-throughput assessment of ENaC-specific inhibition. (**a**) Representative ENaC inhibition with the specific ENaC inhibitor, phenamil (10 µM) or vehicle, DMSO control traces in *opened* HCOs. (**b**) box and whisker plot of ENaC inhibition in *opened* iPSC-derive *opened* non-CF HIOs and HCOs, *opened* CF HIOs, and *opened* H1 HIOs with acute treatment with Amiloride (10 µM or 50 µM) or Phenamil 10 µM relative to DMSO control (**** *p* < 0.0001, *n* = 3 biological replicates, *n* = 4 technical replicates). Each biological replicate = independent organoid passage, technical replicate = a well of a 96-well plate. Box plot depicts the median value and bounds depict IQR ranges with the whiskers defining the minima and maxima values.

**Figure 7 cells-10-03419-f007:**
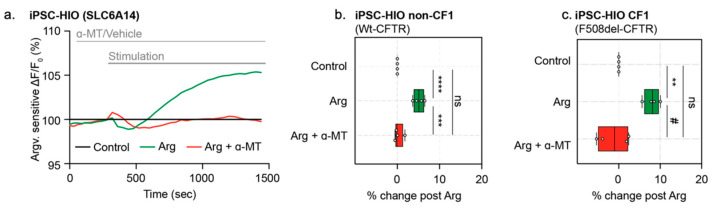
*Opened* iPSC-derived HIOs enable specific measurement of SLC6A14 activity. (**a**) Representative trace of SLC6A14 activity in *opened* non-CF HIOs pre-treated with either vehicle control or inhibitor, α-MT (2 mM). *Opened* HIOs were acutely treated with Arg (1 mM). (**b**) Box and whisker plot of SLC6A14 function with acute treatment of Arg (1 mM) in non-CF HIOs. SLC6A14 activity was inhibited with pre-treatment of specific inhibition α-MT (2 mM) (*** *p* = 0.0001, **** *p* < 0.0001, *n* = 4 biological replicates, *n* = 3 technical replicates). (**c**) Box and whisker plot shows SLC6A14 activity with acute treatment with Arg (1 mM) or in presence of specific inhibitor α-MT (2 mM) in iPSC CF HIOs (** *p* = 0.0080, # *p* = 0.0023, *n* = 3 biological replicates, *n* = 3 technical replicates). Box plot depicts the median value and bounds depict IQR ranges with the whiskers defining the minima and maxima values.

**Table 1 cells-10-03419-t001:** Definition of ESC and iPSC lines used for organoid differentiation.

ESC and iPSC Lines	CFTR Genotype	Description and Source
CF 1	F508del/F508del	iPSC / CFIT Program ^1^
Non-CF 1	Wt/Wt	iPSC / CFIT Program ^1^
Non-CF 2	Wt/Wt	iPSC / CFIT Program ^1^
H1	Wt/Wt	ESC / Cincinnati Children’s Hospital ^2^

^1^ CF Canada-Sick Kids Program for Individualized CF Therapy (CFIT). ^2^ Cincinnati Children’s Hospital Pluripotent Stem Cell Facility (PSCF).

**Table 2 cells-10-03419-t002:** List of primers used for RT-qPCR studies.

Primer Sequence	
*CFTR*	Fwd: 5′-CGGAGTGATAACACAGAAAGT-3′
	Rev: 5′-CAGGAAACTGCTCTATTACAGAC-3′
*αENaC*	Fwd: 5′-TTGACGTCTCCAACTCACCG-3′
	Rev: 5′-GGCAGAGGAGGACAAAGGTC-3′
*SLC6A14*	Fwd: 5′-GCTTGCTGGTTTGTCATCACTCC-3′
	Rev: 5′-TACACCAGCCAAGAGCAACTCC-3′
*EpCAM*	Fwd: 5′-AACACAAGACGACGTGGACA-3′
	Rev: 5′-GCTCTCCGTTCACTCTCAGG-3′
*Villin*	Fwd: 5′-CTGTGATGTCCAGCGACTGT-3′
	Rev: 5′-CTCTCTGGCCCATTCCACTG-3′
*MUC2*	Fwd: 5′-GCTGCTATGTCGAGGACACC-3′
	Rev: 5′-GGGAGGAGTTGGTACACACG-3′
*TBP*	Fwd: 5′-CAAACCCAGAATTGTTCTCCTT-3′
	Rev: 5′-ATGTGGTCTTCCTGAATCCCT-3′

## Data Availability

The data presented in the current study are available upon request to the corresponding author.

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
