# Peer review of "High-Throughput Functional Analysis of CFTR and Other Apically Localized Proteins in iPSC-Derived Human Intestinal Organoids"

_cells, 2021, doi:10.3390/cells10123419_

Round 1

Reviewer 1 Report

Comments to the Author

In their manuscript entitled “High-throughput functional analysis of CFTR and other apically localized proteins in iPSC derived human intestinal organoids,” Sunny Xia et al describe the combination of the methodology Fluorescence Imaging Plate Reader (FLIPR) membrane potential–based assay with 2D cultures originated from iPSC derived human intestinal organoids (HIOs) to measure CFTR, ENaC channel activity and also de sodium and chloride dependent amino acid transporter, SLC6A14, function.

The authors claim that with methodology (that is not novel in this paper) applied to the 2D intestinal cultures called open HIOs can be used in a high throughput manner to measure CFTR, ENaC and SLC A14 function in an indirect manner correlated to the membrane depolarization using a FLIPR blue and a plate reader device.

The authors report here the work done to obtain open HIO and to measure CFTR, ENAC and SLC A14 function using the FLIPR assay based on membrane depolarization. And also, the usage of this methodology to measure correction of CFTR activity using CFTR modulators. This model shows that functional assays can be done in iPSC derived 2D organoids and promises the possibility to increase the thought put to measure CFTR or ENaC rescue, however the capacity of high throughput was not tested in the paper. In summary, this paper brings knowledge that may be considered to publication, however there are several concerns below that must be addressed before this manuscript is ready for publication.

MAJOR CONCERNS:

In figure 1 the authors show in a squeme that the non-CF were produced by gene edited of the CF cells (iPSC cells I deduce) however that is not cleared stated in the text. In materials in methods is stated that CF and non-CF iPSC cells were provided by CFIT. Are all the non-CF iPSC cells and respective organoids derived from gene edited CF iPCS cells? Are the all CF iPSC referred in the paper from a F508del/F508del donor? And all the CF organoids derived from those F508del/F508del iPSC cells? What is the H1 embryonic STEM Cells (ESC) derived organoids? This is important to be clarified in the paper so the reader can understand better which type of organoids are being analysed in which assay. Sometimes the authors stated iPSC-derived non-CF organoids other times iPSC-derived gene edited organoids or even mutation corrected non-CFTR 3D HIOs, are those the same organoids derived from the same gene edited non-CF iPCS cells. For the CF organoids the same type of clarity is needed. In some section the authors refer opened iPSC derived CF intestinal organoids, other times F508del CF organoids. In this paper the CF organoids are only F508del or are there other kind of CF genotype used? Are all non-CF organoids coming from the same CF gene corrected to WT iPSC cells? Please explain this in materials and methods clearly and create an uniform name for CF and non-CF cells, 3D organoids and 2D organoids to make it easier to follow the work done over the paper. Some explanation about the gene edition done to create the non-CF cells should also be given in the materials and methods section.

The organoids CF (F508del/F508del) shown in figure 1 without any stimulation of CFTR (DMSO) already have a lumen, which means that they are already swollen. This is not what is expected at least based on the organoids derived from crypts extracted from rectal biopsies (Dekkers at al Nat Med. 2013 Jul;19(7):939-45), where the CF organoids (F508del) do not have lumen, the swelling (lumen appearing) only occurs upon CFTR rescue with CFTR modulators. The difference between CF and non-CF organoids morphology at baseline (DMSO t0) is not clear in this figure (fig.1). How do the authors justify this swollen phenotype of the F508del organoids at baseline, is CFTR already active? If so, what would that mean for the other assays using these organoids as open organoids. This information is critical to understand all the results in the subsequent CFTR functional assays done in the paper.

MINOR CONCERNS

Material and Methods:

In section 2.4 there is no indication about the number/amount of organoids plated per well of the 96well plate to produce the opened HIOs used for all the functional assays. Was this quantified? That is important since function depends on numbers of cells expressing the channels in analysis in the paper.

In section 2.9 there is no indication about which gene was used as control/reference gene for quantification of qPCR results and how this quantification was done. For the qPCR RNA extraction which type of organoids were used the 3D organoids or 2D open organoids used, how many wells of each per condition?

In the paper there are western blot results using HBE cells (figure2) but there is no info about these cells and the assay done with these cells in the material and methods section.

In section 2.10 and 2.11 the authors refer samples were collected, but which samples, 3D organoids from 24 wells plates or 2D open HIO from 96wells plates. How many wells were collected for the immunofluorescence and for western blot (WB)? Has protein quantification done for the protein lysate of WB? How much total protein has loaded per well in the western blot gel?

Results:

As referred in the major concerns section, it is intriguing why the F508del organoids already have pre-swollen phenotype without any stimulation of CFTR, since swelling should result from CFTR activity and normally in F508del organoids CFTR is not supposed to be functional. The FIS assay is done over 4 hours and the % of swelling is considerable low if we compare it with the results of swelling show by (Dekkers at al Nat Med. 2013 Jul;19(7):939-45) of FIS after 1 hour in organoids derived from stem cells from crypts extracted from rectal organoids of CF patients. How do the authors justify these differences? Is CFTR less expressed in iPSC derived organoids?

In figure 2.B only few cells express CFTR, is this homogenous in different wells? This can affect the measurement of CFTR activity. Was this evaluated? How do the authors control for the amount of cells per wells and CFTR expression? Do the authors quantify the amounts of cells platted in each wells, are the wells confluent? And is CFTR expression similar over the different wells in the plate? How was this taken in consideration? The channel activity depends on the number of cells in the wells and the cells that are expressing the channel in study.  

In figure 3 it is possible to observe stimulation of CFTR by DMSO. Normally DMSO is not a stimulator of CFTR, how do the authors explain this stimulation visible in 3b, 3d and also 3f?

The authors refer that the EC50 of 0.0287µM of forskolin was found in the CFTR function assay on open HIO with FLIPR and they compared it to 0.128µM forskolin referred in Dekkers et al paper. However, their comparison does not make sense since the 0.128µM is not an EC value found by Dekkers et al but the value that gave the best correlation between the organoids results with the CFTR modulators compared to the clinical trials with patients with the same genotype and treated with the same modulators. Since the authors do not compare theirs results to clinical trials results this comparison of values does not seem to make sense. Can the authors explain better what they intended to compare?   

In figure 3 and 4 the same scale in y axis should be used to make the comparison of the AF/Fo results easier.

Like referred in major concerns, it is important to standardize the name of the cultures used, in figure 3 there is H1 (wt-CFTR), non-CF (WT-CFTR) and non-CF2 (wt-CFTR). What are each one specifically ado they derived from different cells? In the figure legend is not clear. 3c corresponds to the result of the figure 3b and g to the average of 3d and 3f or just 3f? in the text it is referred that non-CF1 opened organoids (line 308) but in the figure there is no CF1-opened organoids just non-CF and nonCF2. Probably in figure 3d should be non-CF1.

In section 3.3 in line 345 the authors referred to figure 3b-3d but should be 4b-4d and the same in line 348 should be figure 4d instead of 3d.

In figure 2e and 5a what is the total protein quantity loaded in each well?

Figure 5d is difficult to interpret since the % of change starts on 100% (as no change) and decrease when changes are observed. That is a strange way to show the results. Would not the better to start on 0% change and go higher although it is a negative change in fluorescence?

How do the authors explain that the MDCK cells transfected with the ENaC subunits (figure5C right) show a lower or equal hyperpolarization when compared to to the open HIO (figure6a and 6c). Shouldn’t hese transfected cells express more ENAC then endogenous ENaC expressing 2D organoids? Was ENaC expression (qPCR) compared between these two cultures? This should be done to better understand these results.

In crypts derived organoids no ENaC function was observed (Zomer-van Ommem DD et al 2018), how is this difference in expression in ENaC to the iPSC derived HIOs can be explained?

In line 398, the authors referred that hyperpolarization is show in a sodium dependent manner, but that is not straightforward from the experiment done, is a conclusion and not a result. Maybe if a 2D iPSC ENaC knockout is created then the authors can show that the function is directly dependent on ENaC?

In figure 6, the same scale units should be used in 6a and 6c.

Is there a reason why the inhibitor in the function assay of SLC6A14 (figure 7a) tested in an apart experiment and not added after stimulation, so the inhibition can be seen after the stimulation?

Reviewer 2 Report

In their paper, the authors describe a method to measure the function of CFTR and ENaC, in opened iPSC derived intestinal organoids.

The methodology is quite well described but I have some comments and I raise some issues.

Major comments:

  • My main point is about the significance of the work to the scientific community. Indeed, the contribution of opened organoids for high-throughput screening should be better compared to existing techniques and should be better explained regarding ENaC. The methodology you are proposing is interesting but seems cumbersome to perform and will only be used on platforms. Relevant arguments of interest should be accentuated to favor its dissemination. Knowing that CFTR and ENaC are studied separately, what is the greater value of the technique compared to existing techniques?
  • In the introduction, please explain how the idea originated to use opened The authors should also better explain how their proposed method can add value to other high-throughput techniques. Because ENaC is studied in MDCK
  • In the methods, opened organoids cultures are described. This section should be developed and the authors should explain how they are sure that organoids are really opened. What is the proportion between inside- out and inside -in organoids and is this proportion the same in each experiment?
  • In Fig 1B, both characterization of CF and Wt-CFTR HIOs must be shown.
  • It is written that in Fig. 1C Wt-CFTR HIOs are increased in size after FSK. This is not observed on the figure. Furthermore, CF HIO do not respond to FSK whereas they are treated with VX-509. This has to be clarified.
  • At the opposite of what is written in the text, CF HIO in the presence of FSK, VX-809 and VX-770 are not rescued close to what is observed in non-CF (Fig. 1D). I am even surprised that there is no significant difference between them.
  • Regarding the TR-qPCR experiments, it can be observed in Fig. 2D that CFTR expression is decreased in opened non-CF when compared with CF opened. Could you explain this point?
  • The band C of CFTR is less expressed in non-CF opened organoids than in 3D. This is further clear because CNX varies and has a higher amount in the lane. Please explain what are the consequences regarding opened organoids, compared to 3D.
  • Immunofluorescence (CFTR, ZO-1) should also be performed in CF organoids and added to Fig. 2B. ENaC detection has to be performed in opened organoids of both types.
  • HIOs were obtained from 1 CF and 2 non-CF. I am afraid this is nor representative.

Minor comments:

  • Line 94, it seems to me that there is a sentence without a verb. Despite I am not English native, some sentences seems to be weird to me, like the third sentence in the introduction.
  • In the Methods, swelling images were analyzed by a house made algorithm. Is there a reference to add? Is this algorithm proven and reliable?
  • Fig 3A is truncated.

Round 2

Reviewer 2 Report

The authors responded to all our comments.